# The Role of Amino Acids in Non-Enzymatic Antioxidant Mechanisms in Cancer: A Review

**DOI:** 10.3390/metabo14010028

**Published:** 2023-12-31

**Authors:** Elena I. Dyachenko, Lyudmila V. Bel’skaya

**Affiliations:** Biochemistry Research Laboratory, Omsk State Pedagogical University, Omsk 644099, Russia; pc.krd1@mail.ru

**Keywords:** amino acids, cancer, oxidative stress, non-enzymatic antioxidant system

## Abstract

Currently, the antioxidant properties of amino acids and their role in the physicochemical processes accompanying oxidative stress in cancer remain unclear. Cancer cells are known to extensively uptake amino acids, which are used as an energy source, antioxidant precursors that reduce oxidative stress in cancer, and as regulators of inhibiting or inducing tumor cell-associated gene expression. This review examines nine amino acids (Cys, His, Phe, Met, Trp, Tyr, Pro, Arg, Lys), which play a key role in the non-enzymatic oxidative process in various cancers. Conventionally, these amino acids can be divided into two groups, in one of which the activity increases (Cys, Phe, Met, Pro, Arg, Lys) in cancer, and in the other, it decreases (His, Trp, Tyr). The review examines changes in the metabolism of nine amino acids in eleven types of oncology. We have identified the main nonspecific mechanisms of changes in the metabolic activity of amino acids, and described direct and indirect effects on the redox homeostasis of cells. In the future, this will help to understand better the nature of life of a cancer cell and identify therapeutic targets more effectively.

## 1. Introduction

Recent research in the field of cancer cell metabolism is significantly changing our understanding of metabolic processes such as the redox mechanism, carbon metabolism, epigenetic regulation, and immune response associated with tumorigenesis and metastasis [1,2]. This includes reconsidering the role of amino acids and changes in their composition during oncological processes. It is known that cancer cells intensively absorb amino acids and use them as a source of energy, precursors of antioxidants that reduce the level of oxidative stress in cancer, and also as regulators of inhibition or induction of gene expression associated with the regulation of tumor cell activity [3].

The pathophysiological process begins with a change in the oxidative capacity of proteins, through modification of the structure and activity of protein binding sites with other molecules, which are elements in the cascade reaction of antioxidant activity [4,5,6]. The choice of the oxidative process path and its activity is determined the features of the reactive type like its the activity and specificity. It is deteremined by the cell type, state of cell differentiation, state of the extracellular environment, level of antioxidants and antioxidant enzymes. The activity of the redox process will depend on the specific type of modified protein, and this, in turn, depends on the amino acid composition of the protein. In addition to the above, there is the concept of protein network formation, which explains the coordinated interaction between proteins and the activation of the redox pathway [7,8].

A number of studies on model systems have established that amino acids exhibit the ability to reduce damaging oxidative effects of various natures at different levels of biological organization [9]. These effects probably happen because of the physicochemical properties of individual amino acids, which are connected to their ability to react with reactive oxygen species. For example, the ability of citrulline and Arg to eliminate the superoxide anion radical has been shown, and leads to normalization of the functioning of the heart muscle when exposed to oxidizing factors [10,11]. It has been established that proline is an effective scavenger of singlet oxygen and prevents cell death under oxidative stress [12]. The ability of His to intercept peroxyl radicals and prevent the carboxylation of proteins and the formation of protein cross-links has been revealed [13]. It was discovered that a number of amino acids prevent the formation of 8-oxoguanine in DNA by protecting guanine from one-electron oxidation to the guanine radical cation [14]. It is still unclear to what extent the change in amino acid composition is tissue-specific, as well as how certain types of cancer are metabolically dependent on a specific amino acid composition [15].

It is known that free amino acids can influence the production of hydrogen peroxide and hydroxyl radicals in aqueous solutions when exposed to X-ray radiation [9]. According to the literature, the α-carbon atom is an essential part of all amino acids; it is capable of reacting with a hydroxyl radical, which results in the abstraction of a hydrogen atom at the α-carbon atom to form an amino acid radical [16]. In addition, side amino acid residues are also capable of reacting with the hydroxyl radical; therefore, individual amino acids are observed to have different effects on their formation. In this regard, six amino acids that react most actively with hydroxyl radicals and significantly reduce their production can be identified: Cys, His, Phe, Met, Trp, Tyr [9]. The other most common sites of amino acid oxidation in proteins include Pro, Arg, and Lys [16]. It can be assumed that the oxidation of these amino acids occurs partially along a different path, which is not associated with the action of hydroxyl radicals, for example, through superoxide anion radicals and peroxides. Indeed, Arg and Lys are characterized by their interaction with superoxide anion radicals, and Pro very efficiently forms peroxides [17].

We have specifically chosen these amino acids for a more detailed consideration of their role in the non-enzymatic oxidative process that occurs in various types of cancer because of their features. We have identified nine basic amino acids that play a key role in redox processes. These amino acids include Cys, His, Phe, Met, Trp, Tyr, Pro, Arg, and Lys.

## 2. Basic Aspects of Amino Acid Metabolism during Redox Stress in Oncology

In oncological diseases, an imbalance of redox homeostasis occurs [1]. At local levels, cancer cells use amino acids as elements of antioxidant protection against the body’s aggressive response to their growth and development. At the systemic level, oxidative stress increases due to the body’s reaction and the activation of free radicals, which are reactive oxygen species that aim to destroy cancer cells. In addition, the cancer cell itself reprograms the metabolism of amino acids, which leads to the synthesis of by-products, thereby increasing oxidative stress. This allows tumor cells to freely invade adjacent tissues and organs and cause unhindered metastasis to distant organs [1,2,3].

Conventionally, amino acids are commonly thought of as the building blocks of protein synthesis [5]. However, according to the latest data, their role is much broader. They take part in the regulation of protein metabolism and signal transmission, maintain redox homeostasis, act as an energy source along with glucose, take part in the transport of nitrogen and carbon throughout the organs and tissues of the body, and act as neurotransmitters [2,3,4,5,6,7]. The uniqueness of amino acids is due to the presence of a free carboxyl group and a free amino group on the a-carbon atom [3,4,5,6]. Differences in the properties of amino acids are due to differences in side chains. The biosynthesis of amino acids includes several metabolic pathways through which they are synthesized from other precursors [5,6,7,8,9,10,11]. A significant difference between the biosynthesis of amino acids and the biosynthesis of lipids and carbohydrates is the active use of nitrogen [5].

A specific relationship between amino acid biosynthesis and cancer cell metabolism has not been identified yet. This can be explained by the fact that, firstly, not enough data have been collected to allow us to draw conclusions about the specific relationships between amino acid content and cancer metabolism. Secondly, amino acids are molecules that are basic elements in ensuring the normal functioning of the cell. It must be remembered that any pathology leading to an imbalance of homeostasis will affect the uneven metabolism of amino acids. The main issue here is that the aggressiveness of the pathology directly affects the basic elements of cell life, including the amino acid composition. An important reason for studying amino acid metabolism is that there are still not enough data regarding (1) how a cancer cell maintains its energy balance other than the active consumption of glucose; (2) through which mechanisms metastasis occurs, and how they can be regulated; (3) due to which specific abilities the cancer cell bypasses the attack from the immune system; and (4) how exactly the imbalance in the redox system occurs. A deep understanding of what type of cancer causes reprogramming of the amino acid metabolic pathway and what positive effect this brings to the cancer cell will primarily help to identify effective therapeutic targets, and to better understand the nature of the cancer cell.

Our review identified several possible amino acid derivatives that arise during metabolic reprogramming. When the Phe content is imbalanced, phenyllactic and phenylacetic acids accumulate [18]. With an increase in Met content, homocysteine accumulates [19]. Disorders of Trp metabolism lead to the accumulation of kynurenine, kynurenic acid, 3-hydroxykynurenine, anthranilic acid, 3-hydroxyanthranilic acid, picolinic acid and quinolinic acid [20]. An increase in Arg either directly leads to the activation of genes such as epidermal growth factor receptor (EGFR) and tuberous sclerosis complex 2 (TSC2) [21], or excessive accumulation of nitric oxide causes the expression of the vascular endothelial growth factor (VEGF) gene [1]. A decrease in the activity of hydrogen peroxide [1] and a decrease in the content of NAD(P)+/NAD(P)H also allow for the growth of cancer cells [2]. In addition, the formation of interesting compounds such as nitrogen–oxygen–sulfur bridges in proteins, coinciding with an increase in Lys content, also favors the growth and activity of cancer cells [4].

In oncology, the state of redox stress increases, and amino acid metabolism plays a significant role in this. Some of them (Pro, His) directly affect the production of reactive oxygen species, allowing for the action of hydrogen peroxide; others (Met, Cys) increase the production of glutathione, as one of the strong antioxidants that minimizes the amount of reactive oxygen species and neutralizes free radicals in the body. Arginine increases the production of nitric oxide. High Trp content leads to an increase in de novo NAD+ synthesis, which enhances antioxidant activity. It is important to note that the role of amino acids in antioxidant defense is twofold. In the normal state of the body, all of the listed antioxidant agents work as protectors, but in oncology, some of the listed molecules can have a damaging effect. At what stage and under what exact circumstances this change occurs remain to be investigated.

## 3. The Role of Individual Amino Acids in Redox Processes

### 3.1. Arginine

Particular attention is drawn to Arg and its ambiguous role in the tumor cell. On the one hand, an increase in Arg content leads to an increase in the formation of nitric oxide (NO). Nitric oxide is known to be a strong antioxidant [22]. In this case, Arg can be indirectly attributed to antioxidant potential. By acting as an antioxidant, Arg can scavenge O^2−^ and thereby prevent eNOS-mediated production of O^2−^ in the unbound state. In addition to its indirect antioxidant effects [23,24], NO prevents oxidative stress in tissues by interrupting the lipid peroxidation chain reaction through the formation of non-radical new nitrogen-containing lipid adducts [25] and inhibiting the catalytic activity of CYP2E1 [26,27]. Secondly, NO can trigger the expression of antioxidant enzymes and new genes for resistance to nitrosative stress [28]. Finally, NO enhances the antioxidant activity of glutathione (GSH) [29] by producing S-nitrosoglutathione, which is approximately 100 times more potent than GSH [30]. GSH is not only a major antioxidant, but also activates glutathione peroxidase activity, which protects against oxidation and nitration reactions [31].

There are the additional ways in which Arg can modulate cellular signaling pathways [23]. As already mentioned, Arg is a precursor of NO through the action of NO synthase (NOS) [32]. The dual functions of NO as a signaling messenger in cancer have been noted [33]. NO has been considered as an antitumor reagent due to its antioxidant and radical scavenging properties [34]. However, new data show a diverse role of NO in tumorigenesis, including angiogenesis, metastasis, anti-apoptosis, and anti-host immune response [35]. Increased levels of NO and NOS expression are observed in cancer patients, which strongly correlates with VEGF expression, angiogenesis and metastasis [1,2] (Figure 1). Induction of NO and NOS leads to inactivation of tumor suppressors such as p53 and pRb [36,37]. Induction of NOS and cyclooxygenase (COX-2) has also been shown to increase NO and prostaglandin levels, leading to angiogenesis in cancer [38,39,40]. According to scientific research in recent years, a significant concentration of Arg in a cancer cell can also lead to the activation of another metabolic pathway, namely the activation of mTOR1, which contributes to aggressive and invasive tumor growth. This occurs through S-nitrosylation of key molecules involved in cancer induction, such as EGFR and TSC2, which affects the mammalian target of rapamycin (mTOR) pathway [21,41].

Another interesting discovery was made about the proportional change in the Arg level and its decrease and increase in the content of amino acids such as Trp, Cys and histamine, which led to the suppression of oxidative stress. This process may be due to two reasons. The first includes the ability of a tumor cell to inactivate the suppressor protein p53, which contains the maximum amount of Arg. Another is that His, Trp and histamine reduce the level of oxidative stress in the tumor microenvironment, thereby ensuring high survival of the cancer cell [42,43]. Apparently, high concentrations of Arg have contradictory effects on the body, from antioxidant activity to tumorigenesis-stimulating activity, which is associated with the internal ratio of the concentration of Arg itself to other biological products. This is what may determine further determination of the metabolic pathway of this amino acid. This assumption seems reasonable, since the launch of the cascade reaction of the metabolic process is controlled quantitatively, i.e., in the concentration of biological molecules, and qualitatively, i.e., through activation of specific molecules. In the future, it will be necessary to study the dependence of Arg concentration on the type of pathology and other products, which will ultimately determine the metabolic pathway of Arg, and, consequently, its biological effect.

### 3.2. Tryptophan

In recent decades, the role of Trp in pathological processes, ranging from chronic systemic inflammation to cancer, has been actively studied [44,45,46]. Thus, an inverse relationship between Trp content and the risk of developing colorectal cancer has been shown, i.e., in patients with a high Trp content, the risk of developing this pathology is significantly lower. This is due to a number of reasons. Trp is actively metabolized by the microbiota of the large intestine, resulting in the formation of end products in the form of indole and indole-3-propionylic acid. These metabolites cover the surface of enterocytes, exerting a protective effect against direct mechanical irritation and the chemical and biological effects of nutrients and microorganisms, and reduce the permeability and leakiness of enterocytes, which may cause the development of colorectal cancer [47,48,49,50,51]. Additionally, Trp is the third source of de novo synthesis of NAD+ (NAD+ is also synthesized from nicotinic acid and nicotinamide mononucleotide), which, in turn, changes via the pentose phosphate pathway [52,53] into NADPH-, which is actively involved in antioxidant protection and regulation intercellular signaling through the NADH oxidase (NOX) system (Figure 2) [54,55].

The indirect role of Trp in the redox process was mentioned in a study of Trp metabolism through the kynurenine pathway and its role in the immunobiology of breast cancer [56]. Trp has three possible metabolic pathways: indole, serotonin, and kynurenine. Particular attention was paid to the kynurenine pathway, especially its metabolites: kynurenine, kynurenic acid, 3-hydroxykynurenine, anthranilic acid, 3-hydroxyanthranilic acid, picolinic acid, and quinolinic acid [57]. Under certain intracellular conditions or in a certain microenvironment, the listed products of this pathway are capable of generating reactive oxygen species such as hydrogen peroxide and hydroxyl radicals. This is due to the emergence and progressive growth of the tumor, leading to lipid peroxidation [57], DNA damage or mutation [20,58], and cell proliferation [59,60,61,62,63,64,65,66,67,68,69].

### 3.3. Histidine

Interesting data on the redox properties of His were obtained by a group of scientists from the University of Nebraska Medical Center, USA. In their work, they demonstrated a connection between pancreatic cancer and His levels. It was found that in pancreatic cancer, the level of His decreases, which is a protective mechanism of the cancer cell, since a significant amount of His increases oxidative stress in the cell due to increased production of hydrogen peroxide [70]. Increased expression of the HAL gene was observed, indicating active His metabolism. In their work, they also referred to previous studies in which changes in the metabolism of cancer cells were observed, but in non-small cell lung cancer. In this case, the patients were initially resistant to antitumor treatment, but after adjusting the chemotherapy with the addition of hydrogen peroxide, His increased its cytotoxic effect (Figure 3) [71].

Another group of scientists studied the effect of His on DNA damage and assessed the cytotoxicity caused by hydrogen peroxide in cultured mammalian cells. They showed that His markedly enhanced the growth and DNA synthesis-inhibiting effects of hydrogen peroxide in cultured Chinese hamster ovary cells. DNA single-strand breaks were also higher in the presence of the amino acid, and in addition, these breaks had a slower rate of repair compared to breaks caused by the oxidant alone. In the presence of His, hydrogen peroxide also caused a double-stranded DNA break, a damage that was undetectable in cells treated with even extremely high concentrations of a single oxidant. The data presented here suggest that His-mediated enhancement of the cytotoxic response of cultured Chinese hamster ovary cells to hydrogen peroxide may be at least partially dependent on the formation of DNA double-strand breaks [71].

### 3.4. Tyrosine

Tyr, like any other amino acid, is a building material for proteins on the one hand, and acts as an alternative source of energy for cell life on the other hand. The main site of Tyr metabolism is the liver. In it, the formation of precursors for gluconeogenesis and ketogenesis occurs, as well as the decomposition of amino acids into secondary metabolites with the participation of five key enzymes. Disturbances in Tyr metabolism lead to the occurrence of pathologies such as Huntington’s disease [72] or esophageal cancer [73,74]. Nguyen et al. conducted a comprehensive cross-platform data study to search for potential therapeutic and preventive biomarkers for hepatocellular carcinoma (HCC) [75]. They found a pathologically reduced level of expression of the genes tyrosine aminotransferase (TAT), heme-dependent peroxin (HDP), homogentisate-1,2-dioxygenase (HTG), Glutathione-S-transferase zeta 1 (GSTZ1), and fumarylacetoacetase (FAH), and a consequent decrease in the translation of these enzymes [76]. A positive correlation has been established between the levels of enzymes such as TAT, HTG, and GSTZ1 and the survival of patients with an established diagnosis of HCC. An explanation for the positive prognosis associated with TAT is given in a study by Fu et al. [77]. The proapoptotic effect of tyrosine aminotransferase was experimentally demonstrated by comparing two cell cultures, one of which was transfected with TAT; the other was transfected with an empty vector. The apoptosis index was assessed before and after treatment with straurosporine. The apoptosis index was higher in cell culture with the TAT vector. The proapoptotic effect was due to changes in mitochondrial membrane potential. The mechanism of apoptosis was triggered as follows: the mitochondrial permeability of the cell increased, which led to an increase in the release of Cytochrome-C into the cytoplasm. Cytochrome-C changed the conformation of Apaf-1, initiating apoptosis through activation of caspase-9, which activated caspase-3 and Poly ADP Ribose Polymerase (PARP) [78]. High expression of GSTZ1 enhanced oxidative phosphorylation and respiratory electron transport, which had a negative effect on the development of HCC. Santacatterina et al. found that limited oxidative phosphorylation activity in hepatocytes is associated with decreased apoptotic cell death and increased cancer development (Figure 4) [79].

Changes in the metabolism of Tyr (and other serum amino acids) are observed in many types of cancer pathology. Much research work has been carried out on serum metabolic profiling using a combination of LC-MS and 1H-NMR, for example, by Zhang et al. [80]. Their study was conducted with several groups: a control group, and patients with esophageal adenocarcinoma, Barrett’s esophagus, and high-grade esophageal dysplasia, respectively. The results of this study showed differences between healthy people at risk and cancer patients. There was a gradual depletion of free amino acids: Val, Leu/Ile, Met, Trp, 5-hydroxytryptophan, Tyr, fatty acids with parallel accumulation of glucose, lactate, carnitine, glutamate, and Lys [81,82,83,84,85].

### 3.5. Methionine

Met plays a significant role in antioxidant processes in oncology [86,87,88,89,90]. According to its physicochemical properties, Met participates in redox reactions, reducing the level of oxidative stress both by stimulating the synthesis of glutathione as the main antioxidant agent and through the formation of methyl sulfoxide (MetO) (Figure 5) [91,92,93,94,95]. The modified form of Met binds reactive oxygen species, thereby suppressing oxidative stress. This effect is most likely to be observed in a generally healthy human body. Moreover, with more significant pathological imbalances, Met can act as one of the reasons for increased oxidative stress [96,97,98].

With an abundant intake of Met from food, the substrate flux through the trans methylation pathway decreases, and that through the trans sulfuration pathway increases. The occurrence of various defects in this pathway can lead to fasting homocysteine levels. Abundant accumulation of this amino acid increases oxidative stress, reduces the availability of nitric oxide, and enhances inflammatory reactions [19,99]. In experimental studies, restricted Met intake has been demonstrated to cause a reduction in oxidative stress. Apparently, during oncological processes, a serious biochemical imbalance and modification of metabolic processes occurs, leading to the depletion of the Met enzymatic system and the accumulation of a large number of intermediate products that have an altering effect [100,101,102,103,104,105,106].

### 3.6. Cysteine

In various histological types of cancer, a nonspecific increase in total Cys content occurs. This amino acid is involved in the generation of glutathione (a powerful antioxidant molecule) and sulfur-containing products such as taurine, biotin, coenzyme A, and hydrogen sulfide through various metabolic pathways [107,108,109,110]. Cys is found mainly in the extracellular space in the form of cystine and in small amounts in the form of Cys, which subsequently takes part in non-enzymatic antioxidant mechanisms. To better adapt to oxidative stress, tumor cells interact with their microenvironment through the lymphatic network, fibroblasts, and immune cells, thus stimulating the transport of extracellular cystine into the intracellular space and converting cystine to Cys, which in turn supports the synthesis and secretion of glutathione [111] in the tumor microenvironment [112,113,114,115,116,117]. Recent studies have reported that Cys potentiates the reprogramming of carbon and sulfur metabolism, which underlies the adaptation of ovarian cancer cells to a hypoxic microenvironment through the production of hydrogen sulfide, which enhances the uptake of glucose by cancer cells and facilitates the production of reactive oxygen species [118,119,120,121,122].

An increase in Cys content has been demonstrated in several recent studies in different types of cancer: epithelial ovarian cancer [123], colorectal cancer, prostate cancer, papillary thyroid cancer, breast cancer, and chronic myelogenous leukemia [124,125]. The increase in Cys content in such different types of cancer once again emphasizes the nonspecificity of the process of metabolic reprogramming, and more so characterizes the general fundamental processes underlying the life of a cancer cell, which are still far from being fully understood.

### 3.7. Proline

Pro is a non-essential amino acid. It makes a significant contribution to the formation of protein structures and the determination of protein functions, and also maintains cellular redox homeostasis [126]. Intracellular catabolism of Pro can generate both reactive oxygen species and adenosine triphosphate (ATP). Recent studies have demonstrated that Pro biosynthesis and catabolism are important processes in pathological conditions. This is due not only to the fact that this amino acid is responsible for protein synthesis, but also to the fact that it regulates the redox balance and is a source of energy. Pro is synthesized by the cells of the body and is also supplied with food. The main site of Pro metabolism is the small intestine. About 60% of free hydroxyproline and Pro are released into the bloodstream [3]. Recent studies have demonstrated that there is a significant increase in hydroxyproline concentrations in liver fibrosis and bone tumors [127]. In cachexia [128], carcinogenesis, and diabetes, Pro increases [129]. Increased Pro content in the cerebrospinal fluid and vascular bed of the brain can be used as markers of pathological processes, including oncological ones. Some amino acids, including Pro, due to their large molecular weight, are not able to overcome the blood–brain barrier and enter the brain and spinal cord [130]. Only during pathological processes, when there is a violation of the permeability of the vascular wall, can molecules that previously could not penetrate the brain area in a normal physiological state accumulate in this area, thus affecting the metabolism of the central nervous system [131,132,133].

The enzyme prolineoxidase (PRODH), which is involved in Pro metabolism, can be used as a marker of the oncological process. Elia et al. found an increase in the content of this enzyme in breast cancer. During metastasis, collagen degradation, Pro release, and activation of prolineoxidase occur [134]. Interestingly, prolineoxidase, in turn, promotes tumor progression in non-small cell lung cancer. In this case, the enzyme acts as a chromatin remodeler and lymphoid-specific helicase [135].

Separately, it is worth highlighting the role of Pro in redox homeostasis (Figure 6) [136]. In chronic diseases and oncological conditions [137], a local increase in Pro content occurs [138]. Such pathological conditions are characterized by tissue hypoxia. During inflammation, cytokines are released and stimulate the activation of myofibroblasts and other fibroblast subtypes [139], which are specialized in the deposition of extracellular matrix, including collagen. Collagen consists of 30% prolyl residues. The state of fibrosis during inflammation and cancer requires increased consumption of Pro; therefore, there is an increase in the consumption of exogenous forms of this amino acid and activation of the synthesis of endogenous forms. This can be described as an indirect effect on the redox balance. It is also known that Pro can directly influence redox homeostasis through the removal of reactive oxygen species [140].

The availability of Pro at the systemic and local level affects the balance of proteins [141], nucleotides and cellular NAD(P)+/NAD(P)H [142,143]. When the listed components of homeostasis are disrupted at the level of one cell or group of cells, the local availability of Pro is disrupted, thereby affecting nearby cells. An altered pathological microenvironment is formed.

### 3.8. Lysine

Lys and its derivatives can be used as biological markers of oxidative stress [144]. Lys, upon oxidation, forms allysine (adipic semialdehyde), a protein carbonyl [145,146]. Additionally, von Pappenheim et al. [147] demonstrated the role of Lys and Cys as a redox switch in nitrogen–oxygen–sulfur (NOS) covalent bridge proteins [148]. In most proteins, the NOS switch contains Lys, which is involved in enzymatic catalysis or binding of substrates, DNA, or effectors, thus linking the chemical properties of Lys and the biological functions of the redox process. This NOS bridge functions as an allosteric switch. By influencing the allosteric center of a protein through NOS, one can change its structure and the structure of its active center, and regulate enzymatic activity (Figure 7) [4].

The main discovery of this study is that the family of proteins that are responsible for the response to oxidative stress are involved in the transmission of redox signals; maintain homeostatic balance; are involved in DNA repair, transcription, and translation of ribosomal proteins and degradation proteins [5,6]; and are also involved in signal transduction and biosynthesis of redox-sensitive cofactors contain the amino acids Cys and Lys with great catalytic potential and NOS bridges. With an increase in the content of reactive oxygen species, NOS bridges are formed, which bind to cysteine or lysine, causing their mutations in the allosteric center, thereby changing the structure and function of proteins [149]. These bridges and switches have been found in proteins in diseases such as Alzheimer’s disease, Parkinson’s disease, obesity, autoimmune diseases, cancer, etc. [144].

### 3.9. Phenylalanine

Schulpis et al. have shown that high levels of Phe damage the structure of DNA, proteins, lipids, and also lead to a decrease in antioxidant protection in the body [150,151]. Using the example of patients with phenylketonuria, damage to proteins and lipids was described by recording an increase in products such as carbonyl, oxidized sulfhydryl, the content of active forms of thiobarbituric acid, and malondialdehyde [18]. It was found that Phe metabolites, phenyllactic and phenylacetic acids have a significant effect on antioxidant enzymes (Figure 8). An increase in the content of superoxide dismutase in tissues was observed [152,153].

Recently, more and more information has become available on the concentration of Phe in a particular type of cancer using mass spectrometry. Thus, in their study, Cao et al. demonstrated a high content of the amino acids Phe and Tyr in viable tumor areas of non-small cell lung cancer [154]. Another study found an increase in serum Phe and Tyr in hepatocellular cancer (HCC) [155]. It is noteworthy that there are practically no data on the biochemical processes and reasons for the increase in this amino acid in oncological conditions, and on its role in redox homeostasis.

## 4. Comparative Analysis of the Role of Amino Acids in Non-Enzymatic Oxidative Mechanisms

Based on the analyzed information, we came to the conclusion that in serious pathological conditions, including cancer, a nonspecific change in amino acid metabolism occurs. The data obtained cannot be used as specific markers for a particular type of cancer. At the same time, changes in the activity of amino acids and their role in redox homeostasis can help us to better understand the pathophysiology of the oncological process, and to identify more effective therapeutic targets.

Table 1 summarizes the changes in the activity of the nine most studied amino acids in relation to redox imbalance in cancer. It has been shown that amino acids can be divided conditionally into two groups: those that decrease in concentration in oncology (Tyr, Trp, His), and those that increase (Arg, Phe, Met, Cys, Pro, Lys).

At the same time, for none of the amino acids considered, multidirectional changes in concentration were shown in different types of cancer, which suggests a similar mechanism of involvement in redox processes regardless of the type of cancer (Table 1).

Table 2 provides a summary of the major amino acid metabolic changes in cancer. Each group is characterized by certain mechanisms of alteration and protection in oncology.

Thus, in the group of amino acids whose activity decreases in oncology, the main mechanisms of protection, in our opinion, lie in the formation of intermediate metabolic products and activation of enzymes that are responsible for maintaining the homeostatic properties of a healthy cell (Table 2). Another option for protection is to potentiate the cytotoxic effect on cancer cells by interacting with other molecules [5].

Among the group of amino acids whose activity and content increases, two mechanisms of action can also be distinguished. The first is the reprogramming of amino acid metabolism, which results in the accumulation of intermediate metabolic products that disrupt the body’s antioxidant defenses. The second is a direct effect on the activity of genes and allosteric centers of proteins, which changes the activity of the enzymatic system.

There is not enough information about changes in metabolism when the redox balance is disturbed both in chronic diseases and in oncology for all amino acids discussed in this review. It seems clear that the role of Phe in maintaining redox homeostasis is significant, but has not been adequately studied. A significant increase in this amino acid leads to the accumulation of intermediate products in the form of acids, which significantly affect redox processes. To date, amino acids and their changes are considered exclusively as a descriptive aspect of the disease, as one of many signs of pathology, and no attention is paid to the basic issues of the nature of this change. The answer lies in a targeted study of the physicochemical and biochemical properties of Phe and its metabolic network with other agents in pathology.

Amino acids such as Pro, His, Trp, Tyr, Cys and Arg seem interesting for further study. To date, a sufficient number of studies have been carried out showing that amino acids and their metabolism are much more complex, and their influence on maintaining protein, nucleotide and energy balance is much more significant [1,2,3,4,5,6,7,8,9,10,11]. Moreover, to understand how this or that cell behaves during pathology, it is necessary to review and deepen your knowledge of the field by studying the most basic components of cell activity both in physiological and pathological conditions. We must observe where that preponderance occurs, which processes maintain homeostasis (protecting against the aggressive effects of foreign agents), and which processes have the opposite effect (on the contrary, aggravating the condition of the body). This review has demonstrated that the nature and role of amino acids is controversial. By providing beneficial effects at the systemic level and damaging cells at the local level, an imbalance can occur, and the accumulation of damage at the local level can lead to damage at the systemic level.

## 5. Amino Acids as Therapeutic Targets for Cancer Treatment

Scientists around the world are actively exploring the possibility of regulating amino acid metabolism in oncology [158]. In many cases, therapeutic targeting of tumor cell metabolism produces better results with fewer side effects [121]. For example, therapies targeting the regulation of essential amino acids, such as Met, have beneficial effects. This was demonstrated by scientists using dietary Met restriction in mice and rats, which resulted in an increase in the lifespan of mice and rats [159]. It has long been noted that limiting the supply of amino acids leads to a slowdown in tumor growth [159,160].

Thus, the enzyme arginase is attracting increasing interest as a therapeutic target. There are two types of isoforms of this enzyme, arginase 1 (ARG1) and arginase 2 (ARG2) [161]. Abnormal expression of these enzyme isoforms has been reported to frequently occur in cancers such as breast, gastric, colorectal, and liver cancer [162,163,164,165,166]. In addition, signaling pathways associated with arginase, such as arginase/PI3K/RAC-alpha serine/threonine-protein kinase (AKT)/mTOR, arginase/signal transducer and activator of transcription 3 (STAT3), and arginase/mitogen-activated protein kinase (MAPK), can be influenced [167].

For Lys, the enzyme lysine-specific histone demethylase 1A (KDM1A/LSD1) was discovered. It shows activity in relation to the development, metastasis, progression, and therapy resistance of cancer [168].

Pharmacological modification of Cys residues affects the biological activity of this molecule, thereby demonstrating a therapeutic effect [169]. Thus, it is possible to regulate the entry of cysteine into the cell by blocking the xCT/SLC7A11 transporter [100]. SLC7A11 has been found to be widely expressed in a variety of cancers, and has poor survival rates in patients with breast cancer, prostate cancer, and papillary thyroid carcinoma [170]. In prostate cancer patients, a positive association between xCT expression with invasion and metastasis has been widely found, through an influence on the redox status of the tumor microenvironment [171]. It is also possible to influence GSH peroxidase 4 (GPX4), triggering the mechanism of ferroptosis [172,173,174,175,176,177]. A positive effect in cancer therapy was discovered by suppressing the entry of hydrogen sulfide into cancer cells, which, through activation of GLUT, enhances the uptake of glucose and its glycolysis with the subsequent production of ATP, which is spent on maintaining the life of the cancer cell [178]. Hydrogen sulfide is produced by three enzymes, namely cystathionine β-synthase (CBS), cystathionine γ-lyase (CSE), and 3-mercaptopyruvate sulfatansferase (3MST). CBS knockdown reduced oxygen consumption and ATP production in colon and ovarian cancer cells [179,180]. Suppression of CSE expression in endothelial cells reduced their ability to induce angiogenesis in breast cancer cells [181]. Cys may also have an antitumor effect through its byproduct, taurine. Cysteine dioxygenases catalyze the oxidation of Cys to cysteine sulfinate. Cysteine sulfinic acid decarboxylase catalyzes the reaction, and carboxyl groups are removed to form hypotaurine, which subsequently generates taurine [182]. Taurine has been shown to have cytoprotective functions and maintain cell homeostasis [183]. Increased taurine content has shown an inhibitory effect on the growth of colon cancer [184], lung cancer [185], HCC [186], melanoma [187], and breast cancer [188]. There are a number of other possibilities for influencing Cys, providing an antitumor effect, for example, through protein kinases and phosphatases [189,190,191,192].

An antitumor effect was demonstrated when His was added to HCC tumor cells. It considerably reduces the release of cytokines in response to liver injury [193]. There is a decrease in the expression of tumor markers that are associated with glycolysis (GLUT1 and HK2), inflammation (pSTAT3), angiogenesis (VEGFB and VEGFC), stem cells (CD133), metastasis (snail/slug), and cell migration [194].

Pro synthesized by proline synthase PYCR1 through cGMP-PKG enhances the hallmarks of stem cell carcinoma. Knockdown of proline synthetase PYCR1 showed a significant reduction in breast cancer growth [195]. Another therapeutic target is the enzyme proline dehydrogenase (PRODH) [196]. On the one hand, increased expression of PRODH induces apoptosis [197,198] of breast cancer cells. On the other hand, it stimulates metastasis to the lung. Although there is currently no clear conclusion on the stimulation and inhibition of this enzyme, since its role in cancer metabolism is twofold, further study of the possibility of using proline dehydrogenase as a therapeutic target is very important [199]. Inhibition of Pro synthesis itself may also reduce the synthesis of collagen, which is essential for the growth and development of cancer cells, and the extracellular matrix [200]. It is necessary to find a balance between reducing and completely suppressing the synthesis of extracellular matrix components, since its complete ablation can free tumor cells from the limiting barrier [201,202,203].

To limit the availability of Trp and prevent its use along the kynurene pathway, it is possible to influence the direct transport of tryptophan into the cancer cell, suppress the activity of Trp 2,3-dioxygenase (TDO2) and indoleamine 2,3-dioxygenase 1 (IDO1), and inhibit N′-formylkynurenine formamidase (FAMID), Kyn aminotransferase 1 (KAT1) and kynureninase (KYNU), nicotinamide phosphoribosyltransferase (NMPRT), nicotinamide mononucleotide adenylyltransferase (NMNAT) aryl hydrocarbon receptor (AhR), interleukin 4-induced protein 1 (IL-4I1), poly (ADP-ribose) polymerase (PARPs), and protein acetylaters (SIRTs). All this is a potential target for depriving the cancer cell of its ability to maintain its vital activity [204].

Tyrosine kinase inhibitors have become very widely used in the treatment of various types of cancer. Medicines containing this enzyme, in addition to having a positive effect in the treatment of oncology, also have a number of serious side effects. This forces scientists and physicians to look for new ways to use this approach [205].

Phe has not yet been presented as a therapeutic target for the supervision of cancer patients. This area requires additional study and understanding of possible control points in the life of a cancer cell.

## 6. Conclusions

Research in recent decades has significantly expanded our understanding of the role and functions of amino acids both in health and in cancer. They make a significant contribution to the maintenance of cellular homeostasis, influencing the redox balance and participating in energy metabolism, due to the formation of ATP as a by-product during amino acid metabolism. In addition, amino acid metabolism products are involved in epigenetic regulation and modeling of the immune response, which is associated with tumorigenesis and metastasis. Transporters and transaminases especially attract attention, since they mediate the absorption and synthesis of amino acids, which in the future can be used as markers of metabolic disorders in oncology, and act as a therapeutic target. It is currently not possible to talk about specific mechanisms for increasing or decreasing the activity of certain amino acids, since the topic of amino acids and their metabolism in oncology remains not fully studied.

This area of research is quite promising, as it will allow us to deepen our understanding of the principles of the life of a cancer cell, and will allow us to develop new, more effective therapeutic targets for cancer patients, the number of whom is growing every year (and whose age is becoming younger); it will also allow us to revise and develop a new concept of understanding the properties and functions of amino acids. In the future, there are plans to study changes in metabolism and their impact more deeply, alongside studies of redox homeostasis in specific types of cancer.

## Figures and Tables

**Figure 1 metabolites-14-00028-f001:**
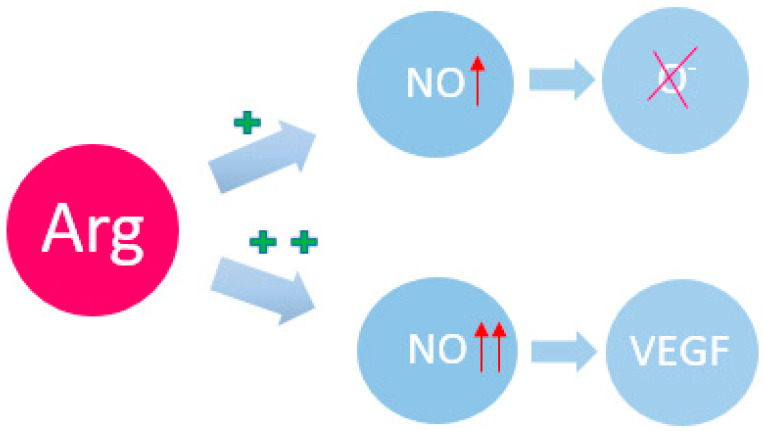
Arg and its participation in redox regulation. NO—nitric oxide; VEGF—vascular endothelial growth factor. The plus sign indicates the induction force; the arrow indicates the magnification force.

**Figure 2 metabolites-14-00028-f002:**
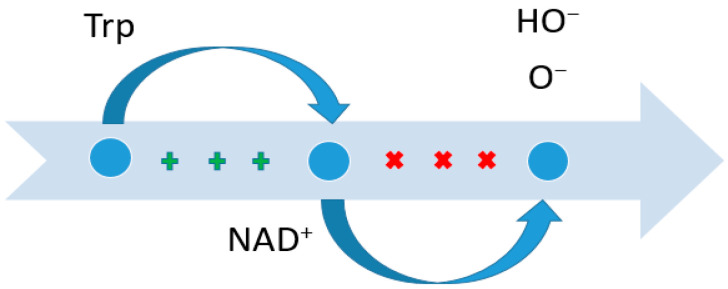
Suppression of the content of certain radicals due to Trp. NAD—nicotinamide adenine dinucleotide. The plus sign indicates the induction force; crosses indicate suppression.

**Figure 3 metabolites-14-00028-f003:**
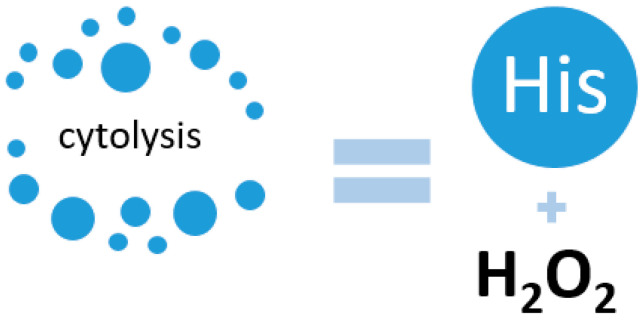
Activation of cell cytolysis due to the interaction of His and hydrogen peroxide.

**Figure 4 metabolites-14-00028-f004:**
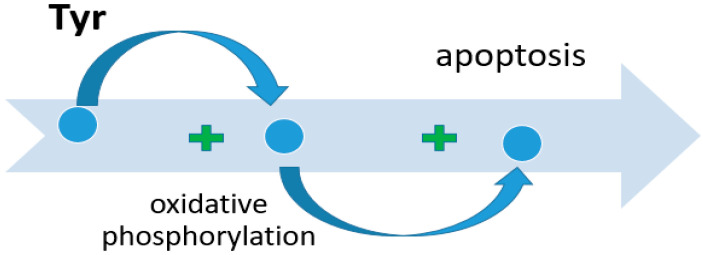
The role of Tyr in the activation of cell apoptosis.

**Figure 5 metabolites-14-00028-f005:**
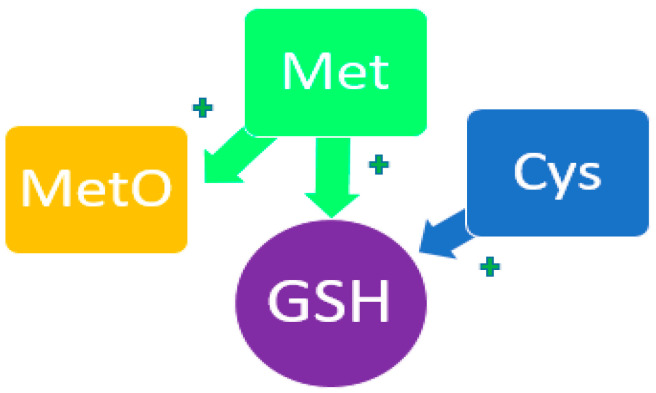
Participation of Met and His in the redox process. MetO—methyl sulfoxide; GSH—glutathione. The plus symbol indicates potentiation of the synthesis of glutathione and methylsulfoxide.

**Figure 6 metabolites-14-00028-f006:**
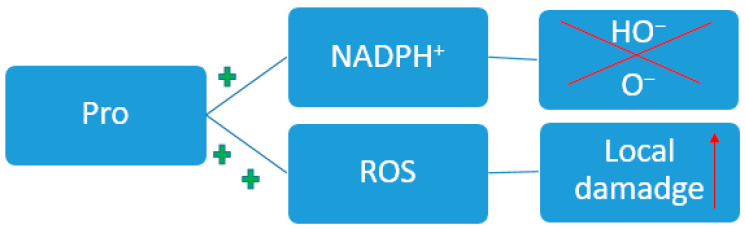
Participation of Pro in the redox process. ROS—Reactive oxygen species; NADPH—nicotinamide adenine dinucleotide phosphate; the plus symbol indicates an increase in the effect.

**Figure 7 metabolites-14-00028-f007:**
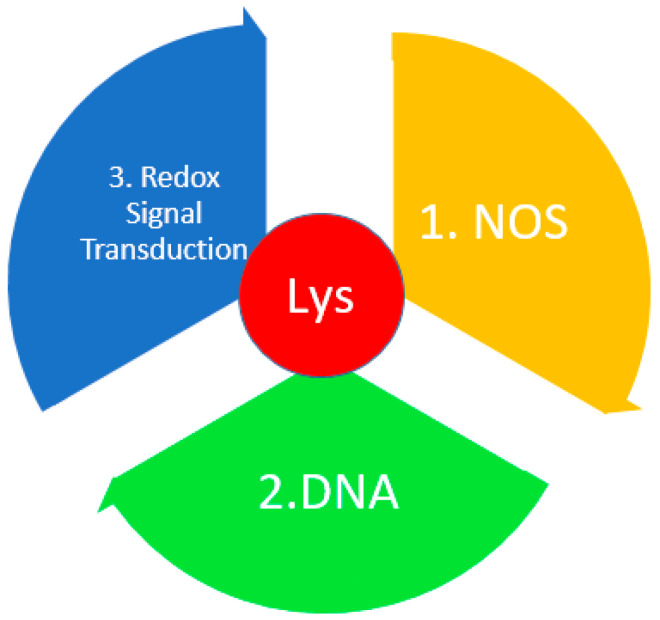
The role of Lys in redox homeostasis. DNA—deoxyribonucleic acid; NADPH—nicotinamide adenine dinucleotide phosphate; NOS—nitrogen–oxygen–sulfur bridge.

**Figure 8 metabolites-14-00028-f008:**
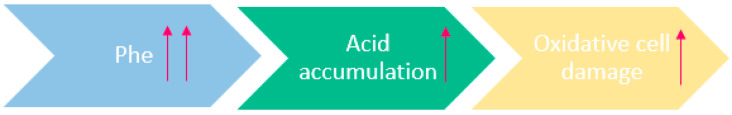
The effect of excess accumulation of Phe on the redox balance.

**Table 1 metabolites-14-00028-t001:** Amino acid concentrations in different types of cancer.

Type of Cancer	Trp	Tyr	His	Arg	Phe	Met	Cys	Pro	Lys
Ovarian Cancer							[106]		
Breast Cancer	[53]	[143]					[124]	[133]	
Liver Cancer		[73]			[155]			[126]	
Lung Cancer		[154]	[133]		[155]			[134]	
Esophageal Cancer		[72]							
Colorectal Cancer	[47,48]						[120]		
Pancreatic Cancer			[68]				[124]		
Prostate Cancer							[117]		
Papillary thyroid Cancer							[124]		
Chronic myeloid leukemia							[124]		
Melanoma		[156]			[157]				

Note. Increases and decreases in the concentration of certain amino acids in various types of oncology. Bright blue color indicates a significant decrease in concentration in oncology, and pale blue color indicates a nonspecific decrease in concentration in oncology; bright pink color indicates a significant increase in a certain type of cancer, and pale pink color indicates nonspecific increase in concentration in oncology.

**Table 2 metabolites-14-00028-t002:** Characteristics of amino acid metabolism in redox imbalance and oncology.

N	Amino Acids	Main Features of Metabolism
1	Arg	(a) Stimulation of angiogenesis through increased expression of the VEGF gene [1,2]. (b) Activation of mTOR1 through influence on EGFR and TSC2 [37,38]. (c) A decrease in arginine content leads to an inversely proportional increase in tryptophan, histamine, and cysteine [39,40].
2	Cys	(a) There is an increase in the production of glutathione, a strong antioxidant agent [110]. (b) Cancer cells take up cystine (the extracellular form), convert it to cysteine, and use it to suppress oxidative stress [106,107,108,109]. (c) In oncology, cancer cells affect cysteine and reprogram the use of sulfur and carbon to combat hypoxic conditions [117,118,119,120,121].
3	His	(a) Histidine potentiates the formation of hydrogen peroxide [68]. (b) Histidine in combination with hydrogen peroxide enhances the cytotoxic effect by double-stranded DNA break [69].
4	Lys	(a) Lys is a redox switch element through interaction with a NOS (nitrogen–oxygen–sulfur) bridge in proteins. It has a direct effect on the allosteric center and an indirect effect on the active center of the protein [4,143,144,145,146,147].
5	Met	(a) Moderate methionine content reduces redox stress through the synthesis of glutathione and methyl sulfoxide, which binds ROS [89,90,91,92,93]. (b) With a significant increase in methionine content, a metabolic shift occurs from transmethylation to transulfuration, which leads to the accumulation of homocysteine [97,98]. (c) Indirectly, through homocysteine, NO availability decreases, oxidative stress increases, and the inflammatory response intensifies [99,100,101,102,103,104,105].
6	Phe	(a) An increase in the formation of intermediate products of phenylalanine metabolism leads to inhibition of enzymes responsible for antioxidant protection [152,153]. (b) Damage to the DNA structure and protein structure occurs, and lipid peroxidation is triggered [149,150].
7	Pro	(a) Proline is involved in the formation of ATP, ROS, and affects protein and nucleotide balance [125,139]. (b) Increased Pro content is observed in oncology and chronic diseases due to release from collagen fibers [135,136,137,138].
8	Trp	(a) An increase in tryptophan reduces the risk of colorectal cancer due to the formation of intermediate products indole andindo-3-propionic acid [44,45,46,47,48]. (b) It is a source of de novo synthesis of NAD+, which determines antioxidant protection and intercellular regulation [49,50,51,52]. (c) When amino acid metabolism is activated along the kynurenine pathway, the risk of developing cancer increases [53,54,55,56].
9	Tyr	(a) The accumulation of the amino acid reduces the risk of developing cancer due to the activation of the enzyme tyrosine aminotransferase (TAT) [73]. (b) Increased proapoptotic index in oncology [75,76].

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
