# Peer review of "The Role of Amino Acids in Non-Enzymatic Antioxidant Mechanisms in Cancer: A Review"

_metabolites, 2023, doi:10.3390/metabo14010028_

Round 1

Reviewer 1 Report

Comments and Suggestions for Authors

The authors reviewed the progresses in the role of amino acids in antioxidant mechanisms in cancer. Since more and more evidence shows the link between amino acids and redox balance in the cells, the review article provided useful information for the field. My concerns are listed below:

1. The rational of the review article should be clearly addressed. For example, the authors stated "We have identified 9 basic amino acids that play a key role in redox processes. These amino acids include: Cys, His, Phe, Met, Trp, Tyr, Pro, Arg, and Lys. ". How the authors identified these 9 amino acids? Why they are more important than others?

2. The authors discussed the role of amino acids in regulation of redox balance in the content of cancer. It is suggestive for the authors to discuss the differences in the role of amino acids between cancer and other cell types. 

3. The authors discussed each amino acid separately. Are there any common mechanisms shared by different amino acids? It is hard for the readers to get general information in the field.

4. The authors should provide several figures to illustrate the underlying mechanisms of amino acids in regulation of redox balance. 

5. Table 1 was not informative, and the information provided in table 1 showed a mild association with the topic.

Comments on the Quality of English Language

The quality of the language is okay.

Reviewer 2 Report

Comments and Suggestions for Authors

The manuscript authored by Elena I. Dyachenko and Lyudmila V. Bel’skaya: ”The Role of Amino Acids in Antioxidant Mechanisms in Cancer: A Review”,  describes the potential role of amino acids in oxidative stress involved in cancer development, focusing on non-enzymatic oxidative process in different types of cancers. The authors have discussed about two separated groups of amino acids, defined by them as: aa with a decreased activity in cancer (Tyr, Trp and His), and aa with increased activity such as Arg, Phe, Met, Cys, Pro, Lys.

The topic of this manuscript is important in the field, but major improvements have to be done in order to be published.

 Major comments:

-          Recently, the role of amino acids in cancer started to be unveiled so it is important that the authors should explain briefly, in one new section, other major aspects related with amino acids and cancer, such as: their biosynthetic pathways, their derivatives that also support cancer growth and the specific relationship between amino acids and cancer metabolism.

-          Also, the antioxidant mechanisms in cancer are not described at all, a briefly presentation of major recent finding of the oxidative stress role in cancer it is necessary in my opinion.

It is important to present at least briefly, major aspects related to oxidative stress-cancer and amino acids biosynthetic pathways, which will help to understand mechanisms described later in text, for each amino acid.

-          Please provide a figure to explain better the relationship between different amino acids and how they influence antioxidant mechanisms in cancer.

-          Table 2 needs a column with references

-          Discussion section should be rewritten! Also, this section has no citations!! For example: Line 348-352 “Thus, in the group of amino acids whose activity decreases in oncology, the main mechanisms of protection consist in the formation of intermediate metabolic products and activation of enzymes that are responsible for maintaining the homeostatic properties of a healthy cell. Another option for protection is to potentiate the cytotoxic effect on cancer cells by interacting with other molecules.”!!! This phrase has no citation and is hard to understand what authors wanted to say.

-          Another example: Line 369 “To date, a sufficient number of studies have been carried out showing that amino acids and their metabolism are much more complex and their influence on maintaining protein, nucleotide and energy balance is much more significant” ???? no citations and what authors wanted to say! Please check all discussion section and add  citations where is needed.

-          The conclusion section should be rewritten also and probably combined with future perspectives.

Please check punctuation and grammar for all manuscript!

Line 79-82: In addition to its indirect antioxidant effect, through the formation of NO, it prevents oxidative stress in tissues [19-20], firstly, by interrupting the chain reaction of lipid peroxidation through the formation of non-radical new nitrogen-containing lipid adducts [21] and inhibiting the catalytic activity of CYP2E1 [22-23]. Please check punctuation of this phrase.

Reviewer 3 Report

Comments and Suggestions for Authors

This review examines nine amino acids (Cys, His, Phe, Met, Trp, Tyr, Pro, Arg, Lys), which play a key role in the non-enzymatic oxidative process in various cancers.

Major revision comments:

[1]   What is the current understanding of the role of amino acids in antioxidant mechanisms in cancer?

[2]   Abstract need to improve.

[3]   Please provide some figures for meachanism.

[4]   How do cancer cells utilize amino acids as both an energy source and antioxidant precursors?

[5]   What is the significance of the nine specific amino acids (Cys, His, Phe, Met, Trp, Tyr, Pro, Arg, Lys) in the non-enzymatic oxidative processes in various cancers?

[6]   Can you elaborate on the dual role of amino acids in regulating tumor cell-associated gene expression in cancer?

[7]   In which group do the amino acids Cys, Phe, Met, Pro, Arg, and Lys fall, and how does their activity change in cancer?

[8]   What are the metabolic changes associated with amino acids His, Trp, and Tyr in cancer, and how does their activity differ from the other group?

[9]   How do these amino acids contribute to the overall redox homeostasis of cells in cancer?

[10]                       Are there specific mechanisms identified for the increased or decreased activity of amino acids in cancer cells?

[11]                       Can you discuss any therapeutic implications or potential interventions based on the understanding of amino acid involvement in oxidative stress in cancer?

[12]                       What gaps or areas of uncertainty still exist in our knowledge regarding the antioxidant properties of amino acids in cancer, and what future research directions should be pursued?

[13]                       Please provide future prospectives.

Comments on the Quality of English Language

This review examines nine amino acids (Cys, His, Phe, Met, Trp, Tyr, Pro, Arg, Lys), which play a key role in the non-enzymatic oxidative process in various cancers.

Major revision comments:

[1]   What is the current understanding of the role of amino acids in antioxidant mechanisms in cancer?

[2]   Abstract need to improve.

[3]   Please provide some figures for meachanism.

[4]   How do cancer cells utilize amino acids as both an energy source and antioxidant precursors?

[5]   What is the significance of the nine specific amino acids (Cys, His, Phe, Met, Trp, Tyr, Pro, Arg, Lys) in the non-enzymatic oxidative processes in various cancers?

[6]   Can you elaborate on the dual role of amino acids in regulating tumor cell-associated gene expression in cancer?

[7]   In which group do the amino acids Cys, Phe, Met, Pro, Arg, and Lys fall, and how does their activity change in cancer?

[8]   What are the metabolic changes associated with amino acids His, Trp, and Tyr in cancer, and how does their activity differ from the other group?

[9]   How do these amino acids contribute to the overall redox homeostasis of cells in cancer?

[10]                       Are there specific mechanisms identified for the increased or decreased activity of amino acids in cancer cells?

[11]                       Can you discuss any therapeutic implications or potential interventions based on the understanding of amino acid involvement in oxidative stress in cancer?

[12]                       What gaps or areas of uncertainty still exist in our knowledge regarding the antioxidant properties of amino acids in cancer, and what future research directions should be pursued?

[13]                       Please provide future prospectives.

Round 2

Reviewer 1 Report

Comments and Suggestions for Authors

No further comments.

Reviewer 2 Report

Comments and Suggestions for Authors

The authors responded satisfactorily to my requirements. The manuscript can be published in this form.

Reviewer 3 Report

Comments and Suggestions for Authors

Requested corrections were completed. 

Comments on the Quality of English Language

Requested corrections were completed.